**Data Availability Statement:** No datasets were generated or analysed during the current study. All

# Psilocybin-assisted massed cognitive processing therapy for chronic posttraumatic stress disorder: Protocol for an open-label pilot feasibility trial

**Shakila Meshkat**[1,2,8‡], **Richard J. Zeifman**[3‡], **Kathleen Stewart**[4‡], **Reinhard Janssen-Aguilar**[1,8], **Wendy Lou**[5], **Rakesh Jetly**[6], **Candice M. Monson**[4], **Venkat Bhat** [1,2,5,6,7,8]*

1 Interventional Psychiatry Program, St. Michael's Hospital, Toronto, Ontario, Canada, 2 Mental Health and Addictions Services, St. Michael's Hospital, Toronto, Ontario, Canada, 3 NYU Center for Psychedelic Medicine, NYU Grossman School of Medicine, NY, NY, United States of America, 4 Department of Psychology, Toronto Metropolitan University, Toronto, Ontario, Canada, 5 Division of Biostatistics, Dalla Lana School of Public Health, University of Toronto, Toronto, Ontario, Canada, 6 The Institute of Mental Health Research, Royal Ottawa Hospital, University of Ottawa, Ontario, Canada, 7 Institute of Medical Science, University of Toronto, Toronto, Ontario, Canada, 8 Department of Psychiatry, University of Toronto, Toronto, Ontario, Canada

‡ SM, RJZ and KS are contributed equally to this work as co-first authors.
* venkat.bhat@utoronto.ca

## Abstract

### Background

Posttraumatic stress disorder (PTSD) affects 3.9% of the general population. While massed cognitive processing therapy (CPT) has demonstrated efficacy in treating chronic PTSD, a substantial proportion of patients still continue to meet PTSD criteria after treatment, highlighting the need for novel therapeutic approaches. Preliminary evidence supports the potential therapeutic action of psilocybin to alleviate PTSD symptoms. This open-label pilot study aims to evaluate the feasibility, tolerability, and preliminary efficacy of a single dose 25 mg psilocybin in combination with one week of massed CPT in patients with chronic PTSD.

### Method

Fifteen participants with chronic PTSD will undergo 12 CPT sessions, two psilocybin-related psychotherapy sessions, and one psilocybin dosing session over a 7-days period. The primary outcomes are feasibility and tolerability, which will be measured by recruitment rates, withdrawal, data completion, adherence, number and nature of adverse events. Secondary objectives include assessing the preliminary efficacy of psilocybin-assisted CPT in reducing PTSD severity, self-reported treatment outcomes and exploring putative mechanisms of change. Participants will be monitored weekly for 12 weeks post-treatment and passive data relevant to mental health and well-being will be collected using a wearable device.

### Discussion

This trial will generate important preliminary data on the use of psilocybin-assisted CPT for treating PTSD. The findings will guide the design of a multi-site, large-scale randomized

relevant data from this study will be made available upon study completion.

**Funding:** The author(s) received no specific funding for this work.

**Competing interests:** I have read the journal's policy and the authors of this manuscript have the following competing interests: VB is supported by an Academic Scholar Award from the University of Toronto Department of Psychiatry and has received research support from the Canadian Institutes of Health Research, Brain & Behavior Foundation, Ontario Ministry of Health Innovation Funds, Royal College of Physicians and Surgeons of Canada, Department of National Defense (Government of Canada), New Frontiers in Research Fund, Associated Medical Services Inc. Healthcare, American Foundation for Suicide Prevention, Roche Canada, Novartis, and Eisai. RJ is the CMO of Mydecine Innovation Group.

control trial to more rigorously assess the efficacy of this intervention. De-identified data from this study will be available upon request after publication of the results. This study represents a promising and innovative approach to PTSD treatment, potentially offering an alternative therapeutic option for individuals unresponsive to conventional therapies.

## Trial registration

ClinicalTrials.gov NCT06386003.

## 1. Introduction

Posttraumatic stress disorder (PTSD) is a severe psychiatric disorder that may develop as a result of a severely stressful experience, such as interpersonal violence, combat, life-threatening accident, or natural disaster [1]. PTSD is associated with significant distress, functional impairment, health burden, and economic cost [2]. PTSD affects approximately 3.9% of the general population [1, 2]. PTSD is classified into two forms based on its duration: acute PTSD and chronic PTSD. Acute PTSD occurs when symptoms last less than three months; chronic PTSD occurs when symptoms last more than three months [3, 4].

Chronic PTSD is particularly challenging as it tends to produce persistent, severe symptoms that can profoundly disrupt a person's daily life and overall functioning [4]. Although guidelines for treating chronic PTSD are well-established, first-line treatments such as antidepressants and psychotherapy often show limited efficacy and can present tolerability challenges for a subset of patients [2]. Cognitive Processing Therapy (CPT) has established efficacy as a treatment for PTSD, with recent advancements in the delivery of massed CPT, involving accelerated session schedules (e.g., completing all 12 sessions over 7 days) [5]. Although gold-standard treatment approaches including massed CPT appear to attenuate PTSD-related symptoms, dropout rates associated with these treatments are approximately one in every five persons, with up to half of individuals deemed non responsive to treatment [1, 6, 7] Accordingly, there is a need for novel adaptations of PTSD treatments that may help to accelerate mechanisms of change and optimize treatment outcomes.

Psilocybin is a 5-hydroxytryptaminergic psychedelic and a non-selective serotonin 2A receptor agonist that has a range of effects on emotion, perception, and cognition [8]. Existing literature emphasizes the therapeutic potential of psilocybin when administered in controlled, optimal conditions [8, 9]. It exhibits transdiagnostic potential due to its rapid onset of action (i.e., within one day of dosing) along with large effect sizes, and high response and remission rates across a variety of psychiatric disorders [10]. Although no studies have directly evaluated psilocybin for PTSD, several lines of evidence suggest its potential in treating the disorder [11]. For instance, in an open-label trial of individuals with long-term AIDS-related demoralization, psilocybin therapy was associated with large reductions in PTSD symptoms post-treatment [12]. Moreover, mechanisms underlying psilocybin therapy—such as reductions in avoidance behavior, fear extinction in animal models, and decreased amygdala activity—align closely with those critical to effective PTSD treatments [13, 14].

Considering these preliminary findings and psilocybin's mechanism of action, psilocybin-assisted CPT for PTSD may be a rapid treatment that optimizes treatment for individuals with PTSD. However, no clinical trials to date have specifically investigated psilocybin, either alone or in combination with CPT, for the treatment of PTSD. To address this knowledge gap, we

plan to conduct an open-label pilot trial to evaluate the feasibility, tolerability, safety and pre-
liminary efficacy of psilocybin-assisted CPT in patients with chronic PTSD.

## 2. Method and analysis

### 2.1. Objectives

The primary objective of this clinical trial is to evaluate the feasibility and tolerability of psilo-
cybin-assisted CPT in adults with chronic PTSD. Secondary objectives include assessing the
efficacy of psilocybin-assisted CPT at point-of-care. Reduction in clinician-rated PTSD sever-
ity, self-reported treatment outcomes and putative mechanisms of change will be assessed. We
hypothesize that both clinician-rated and self-reported PTSD severity will significantly
decrease following treatment. As a exploratory objective, this study incorporate the use of a
digital platform to evaluate the effect of psilocybin-assisted CPT on digital physiological pas-
sive data collected through the use of a wearable device (Oura Ring) and to create a personal
digital phenotype profile (pDPP) based on the wearable and clinical assessment data. The
safety objectives of the study are to monitor and assure the safety of participants during study
participation by assessing severity, incidence and frequency of physiological effects, psycholog-
ical distress, adverse events (AEs), treatment emergent AEs (TEAEs), AEs of special interest
(AESIs), serious AEs (SAEs), medical events, concomitant medication use, and suicidal idea-
tion and behavior.

   The authors assert that all procedures contributing to this work comply with the ethical
standards of the relevant national and institutional committees on human experimentation
and with the Helsinki Declaration of 1975, as revised in 2013 [15]. All procedures involving
human participants were approved by the Unity Health Toronto REB (approval number #23–
230, protocol version #5, approved on 16 April 2024). The trial was registered with Clinical-
trials.gov (identifier: NCT06386003). We will communicate to relevant parties, such as the
Research Ethics Board, Health Canada, and trial participants for protocol modifications.

### 2.2. Setting

This study will take place in the Interventional Psychiatry Program at St. Michael's Hospital,
Unity Health in Toronto, Ontario, Canada.

### 2.3. Design

We propose to conduct an open-label trial in which 15 individuals with chronic PTSD receive
1 week of massed CPT combined with a single dose of psilocybin (25 mg). The primary end-
point will be one week post-treatment (week 2). Participants will use a commercially available
wearable device to collect passive data that is relevant to mental health and well-being, such as
physiological signals, sleep, and activity patterns. As this is a single-group, open-label study,
there is no randomization or blinding. The Consolidated Standards of Reporting Trials (CON-
SORT) flow diagram is indicated in Fig 1. An overview of all study measures and activities can
be found in Figs 2 and 3.

### 2.4. Participants and recruitment

   **2.4.1. Recruitment.**   We anticipate consenting over 100 participants, as some may not
meet eligibility criteria during the in-person screening process, in order to successfully enroll
15 individuals for the trial. These participants will undergo psilocybin administration, CPT
sessions, and complete the final follow-up evaluation. Participants will be recruited from the
Interventional Psychiatry Program at St. Michael's Hospital and referred from the

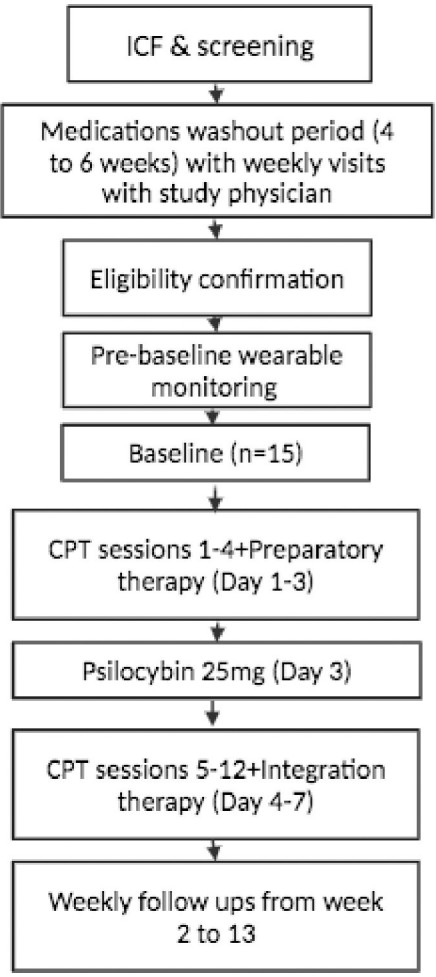

**Fig 1. CONSORT flow diagram.**

Investigating Methods to Prevent, Assess and Care for Trauma (Impact) Laboratory at Toronto Metropolitan University. This study will start recruitment in August 2024.

**2.4.2. Screening.** Participants who express interest to participate in the study will be contacted to perform a brief pre-study screening. During this pre-screening session, the study research coordinator will provide an overview of psilocybin-assisted CPT, explaining its potential benefits and any possible impacts of participation. Eligibility will be assessed through both screening and eligibility confirmation visits, which will be scheduled during this pre-screening call. Pre-screening will involve a secure online questionnaire to determine whether participants meet the major inclusion and exclusion criteria. Participants who meet all study criteria, will be invited to the Interventional Psychiatry Program at St. Michael's Hospital for an in-person screening. If a potential participant is currently taking psychiatric medications, they will be required to taper off these medications under the supervision of the study physician. Written informed consent will be obtained during the in-person screening visit.

**2.4.3. Inclusion/exclusion criteria.** Individuals eligible to be enrolled into this protocol are participants who: (1) Meet Diagnostic and Statistical Manual-5th edition (DSM-5) criteria for current PTSD with a duration of 6 months or longer; (2) Have a Clinician-Administered PTSD Scale for DSM-5 (CAPS-5) score of 50 or higher; (3) Aged 18–80, inclusive; (4) Are

| Study phase | S1 | Pre-baseline wearable monitoring (Oura)** | | | B2 | Treatment phase | | | | | | | PT | Follow-up | | | | | | | | | | | |
|---|---|---|---|---|---|---|---|---|---|---|---|---|---|---|---|---|---|---|---|---|---|---|---|---|---|---|
| Assessment days | | | | | | D1 | D2 | D3 | D4 | D5 | D6 | D7 | | W2 | W3 | W4 | W5 | W6 | W7 | W8 | W9 | W10 | W11 | W12 | W13 |
| **Enrollment** | | | | | | | | | | | | | | | | | | | | | | | | | |
| Eligibility screen, ICF | X | | | | | | | | | | | | | | | | | | | | | | | | |
| MINI, CAPS-5, PCL-5, C-SSRS | X | | | | | | | | | | | | | | | | | | | | | | | | |
| DEMO, Ph/E, vitals, urine test | X | | | | | | | | | | | | | | | | | | | | | | | | |
| Blood test, ECG, MSF | X | | | | | | | | | | | | | | | | | | | | | | | | |
| Contraceptive method review, Pregnancy screening | X | | | | | | | | | | | | | | | | | | | | | | | | |
| **Interventions** | | | | | | | | | | | | | | | | | | | | | | | | | |
| Psilocybin | | | | | | | | X | | | | | | | | | | | | | | | | | |
| CPT (x2) | | | | | | X | X | | X | X | X | X | | | | | | | | | | | | | |
| Preparatory session | | | | | | | X | | | | | | | | | | | | | | | | | | |
| Integration session | | | | | | | | | X | | | | | | | | | | | | | | | | |
| **Clinical assessments** | | | | | | | | | | | | | | | | | | | | | | | | | |
| CAPS-5 | X | | | | X | | | | | | | | | X | X | | | | | | | | | | X |
| PCL-5 | X | X | X | X | X | X | X | X | X | X | X | X | X | X | X | X | X | X | X | X | X | X | X | X | X |
| PHQ-9, GAD-7, WHO-5 | | X | X | X | X | X | X | X | X | X | X | X | X | X | X | X | X | X | X | X | X | X | X | X | X |
| WAI-SF | | | | | | X | X | X | X | X | X | X | | | | | | | | | | | | | |
| DES-II, PSQI, QRI, IPF | | | | | X | | | | | | | | | X | X | | X | | | | | | | | X |
| PMBS, BEAQ, SCS-SF, PSQI | | | | | X | | | | | | | | | X | X | | X | | | | | | | | X |
| MPFI-24, DERS-SF | | | | | X | | | | | | | | | X | X | | X | | | | | | | | X |
| PIQ, EBI, ASC, APEQ | | | | | | | | X | | | | | | | | | | | | | | | | | |
| Oura Ring | | X | X | X | X | X | X | X | X | X | X | X | X | X | X | X | X | X | X | X | X | X | X | X | X |
| Qualitative survey | | | | | | | | | | | | | | | | | X | | | | | | | | X |
| AE log, C-SSRS, CONMED | | | | | | X | X | X | X | X | X | X | X | X | X | X | X | X | X | X | X | X | X | X | X |

**Fig 2. Schedule of enrolment, interventions, and assessments. Abbreviations:** CAPS-5: Clinician-Administered PTSD Scale for DSM-5, PCL-5: PTSD Checklist-5, PHQ-9: Patient Health Questionnaire-9, DES-II: Dissociative Experiences Scale II, PSQI: Pittsburgh Sleep Quality Index, QRI: Quality of relationships inventory, IPF: Inventory of psychosocial functioning, PMBS: Posttraumatic Maladaptive Beliefs Scale, BEAQ: Brief Experiential Avoidance Questionnaire, MPFI-24: 24 items Multidimensional Psychological Flexibility Inventory, WAI-SF: Working Alliance Inventory- Short Form, DERS-SF: Difficulties in Emotion Regulation Scale-Short Form, SCS-SF: Self-compassion scale short form, PIQ: Psychological Insight Questionnaire, EBI: Emotional Breakthrough Inventory, ASC: Altered States of Consciousness Rating Scale, APEQ: Acceptance/Avoidance-Promoting Experiences Questionnaire, GAD-7: Generalized Anxiety Disorder Scale, 7-item, CTQ-SF: Childhood Trauma Questionnaire-short form, WHO-5: 5-item World Health Organization Well-Being Index, CPT: Cognitive Processing Therapy, MINI: Mini International Neuropsychiatric Interview.

willing to refrain from taking any psychiatric medications during the study period; (5) Agree that, for one week preceding the psilocybin session, they will refrain from: a) taking any herbal supplement; b) taking any nonprescription medications unless given prior approval from the research team); c) taking any prescription medications; 6. Refrain from the use of any psycho-active drug, with the exception of caffeine or nicotine, within 24 hours of the psilocybin

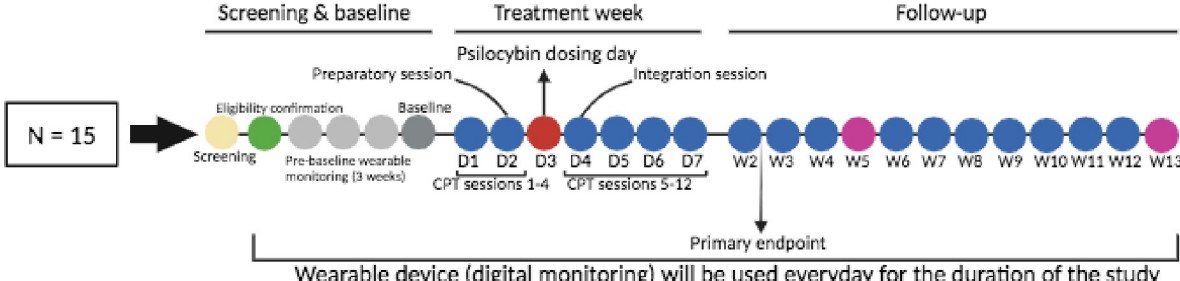

**Fig 3. Study design with timepoints.**

session; 7. Agree not to use caffeine or nicotine for 2 hours before and 6 hours after the dose of psilocybin; 8. Agree not to use alcohol for 14 hours before and 24 hours after the dose of psilocybin; 9. Agree not to use benzodiazepines, hypnotics, and mood stabilizers for 24 hours before and 12 hours after the dose of psilocybin; 10. Agree not to use steroids for 2 weeks before and 2 weeks after the dose of psilocybin; 11. Agree not to use s-Adenosyl methionine, 5-Hydroxy-tryptophan, St. John's Wort for 1 week before the dose of psilocybin; 12. Agree not to use cannabis for 1 week before and 1 week after the dose of psilocybin.

Individuals not eligible to be enrolled into this protocol are those who: 1. Are pregnant or nursing, or are women of child bearing potential who are not practicing an effective means of birth control. 2. Have a history of or a current primary diagnosis of psychotic disorder, schizophrenia, delusional disorder, borderline personality disorder, paranoid personality disorder, schizoaffective disorder, bipolar disorder or, dissociative identity disorder; 3. Have evidence or history of coronary artery disease or cerebral or peripheral vascular disease, hepatic disease with abnormal liver enzymes, or any other medical disorder judged by the investigator to significantly increase the risk of psilocybin administration; 4. Have hypertension using the standard criteria of the American Heart Association (values of 140/90 or higher assessed on three separate occasions; 5. History of seizure disorder; 6. Uncontrolled insulin-dependent diabetes; 7. Recent stroke, intracranial or subarachnoid hemorrhage (< 1 year from signing of informed consent form [ICF]), recent myocardial infarction (< 1 year from signing of ICF), clinically significant arrhythmia (< 1 year from signing of ICF); 8. Have liver disease with the exception of asymptomatic subjects with Hepatitis C who have previously undergone evaluation and successful treatment; 9. Lifetime history of substance-induced psychosis; 10. Lifetime history of substance use disorder with a hallucinogen; 11. History of alcohol use disorder in the past 3 months; 12. Weigh less than 48 kg; 13. Would present a serious suicide risk; 14. Have used psilocybin more than 10 times within the last 10 years or at least once within 12 months of the psilocybin session; 15. Require ongoing concomitant therapy with a psychiatric drug; 16. Active substance abuse or dependence for any substance save caffeine or nicotine in the past 3 months; 17. Positive urine drug screen for illicit drugs or drugs of abuse at screening; 18. Current enrolment in any investigational drug or device study or participation in such within 30 days of screening; 19. Any other clinically significant cardiovascular, pulmonary, gastrointestinal, hepatic, renal or any other major concurrent illness that, in the opinion of the investigator, may interfere with the interpretation of the study results or constitute a health risk for the participant if he/she takes part in the study; 20. Have any current problem or a history of substance abuse which, in the opinion of the investigator or medical monitor, might interfere with participation in the protocol; 21. First degree family history of bipolar I disorder, schizophrenia or any psychotic disorders, including bipolar disorder with psychotic features.

## 2.5. Treatment

Participants will receive a total of 12 CPT sessions, 2 psilocybin-related psychotherapy sessions, and one psilocybin dosing session delivered over a 7-days period.

**2.5.1. Psilocybin dosing session.** Participants will receive a 25 mg dose of psilocybin on day 3 of the treatment week. The study coordinator and the therapist with whom they have been completing CPT sessions will be present on site for the session. The participant should have a light breakfast at least two hours before receiving psilocybin. Under the therapist's direct supervision, participants will swallow the 25 mg of psilocybin. The dosing session will last between 6 to 8 hours. Psilocybin effects typically begin within 20–30 minutes, peaking at 90 to 120 minutes post-administration, and generally subside after 5 to 6 hours. To minimize distraction and interruption, participants will lie down on a couch in a dimly lit room after

taking the psilocybin. To encourage inward reflection, the participant will listen to a pre-programmed selection of music during the session and will be provided an eye mask. A therapist specialized in providing reassurance and a safe environment will accompany the participant until the effects of the dose have passed. A physician will also be on-site to address any emergencies. Additionally, the study coordinator will be present during the session to support the therapist if necessary. Participants are required to remain on-site for the entire duration of the session. They will be encouraged to stay lying down unless they need to communicate discomfort, seek support, or use the restroom. The therapist will check in on the participant's well-being every 1 to 2 hours post-dosing. Approximately 5 to 6 hours after dosing, the therapist will review the psilocybin experience with the participant. Once the therapist determines that the acute effects have subsided—typically between 6 and 8 hours after administration—the participant will be discharged. They will leave the study setting in the care of a responsible individual who will accompany them home and monitor their well-being for 24 hours post-dose. Before leaving, participants will complete pre-dose and post-dose assessments with a research coordinator. The post-dose assessment will occur after the acute effects of psilocybin have resolved. Following the session, participants will also fill out paper or computer-based questionnaires, including the Psychological Insight Questionnaire (PIQ), [16] Emotional Breakthrough Inventory (EBI), [17] Altered States of Consciousness Rating Scale (ASC), [18] and the Acceptance/Avoidance-Promoting Experiences Questionnaire (APEQ), [19] to evaluate their acute experiences.

**2.5.2. Psychotherapy sessions.** Cognitive processing therapy is an evidence-based frontline treatment for PTSD, which includes challenging negative and maladaptive trauma-related thoughts, as well as processing of trauma-related emotions [20, 21]. Although originally delivered on a weekly basis, CPT has demonstrated efficacy when delivered virtually in a massed delivery format [22]. On days 1 and 2 of the treatment period, CPT sessions 1 through 4 and the preparatory pre-dosing session will be delivered virtually. These sessions include psychoeducation about PTSD, reviewing the impact the trauma has had on the individual, identifying maladaptive trauma-related beliefs, and beginning cognitive reframing. The participant will have one preparatory session with the same trained therapist to build rapport, assess patient readiness and prepare the participant for the psilocybin session. The preparatory session will be completed on day 2 and will last approximately 45 minutes. On day 3, the psilocybin dosing session (6–8 hours) will be completed in person. During the psilocybin session, the therapist's goal will be directed toward reducing prolonged symptoms of anxiety or agitation. The therapist will be directive during the session when they feel it is necessary to ensure participant safety, but will avoid active coaching or voicing of interpretations. On the day after the dosing session (day 4), the integration session (1 hour) and CPT sessions 5 and 6 will be completed virtually. These sessions include continued cognitive reframing, with an emphasis on challenging blame-related beliefs. During the integration session, participants will discuss their experience during the psilocybin session with the therapist. CPT sessions 7–12 will be completed virtually twice a day until day 7. These sessions focus on PTSD relevant themes (safety, trust, power/control, esteem, and intimacy). Psychotherapy sessions (and the psilocybin dosing session) will be completed by a single therapist who will remain consistent throughout a participant's treatment. CPT sessions are 90-minutes in length.

## 2.6. Wearable device

Throughout the study, participants will use a wearable device that continuously collects longitudinal, passive data to monitor activity, sleep and physiological metrics such as oxygen saturation, heart rate and heart rate variability. Since the Oura Ring is a passive wearable device, it

can be used without obstructing one's daily life. The use of the wearable device will also involve a mobile app, which will be used for data visualization. The wearable device will be evaluated for its feasibility to collect physiological data in naturalistic settings. It will be helpful in tracking individual trends over time, which will potentially provide us with a better understanding of physiological responses related to PTSD and psilocybin. These multivariate models will help us build prediction tools, which could be an important factor in the design of decision support tools related to mental health. We will incorporate sex/gender- and age-based analysis throughout the study to adjust the models as needed.

## 2.7. Follow up

The follow-up period lasts a total of 12 weeks, with assessments occurring weekly after the treatment week. Follow-up evaluations will occur virtually from weeks 2 to 12, with an in-person assessment scheduled for week 13. A narrative, qualitative survey at week 5 and week 13 will evaluate participant experiences with psilocybin-assisted CPT and the wearable device. Participants will return the wearable device at week 13. The interviews will take approximately 45–60 minutes to complete. Additionally, we will also use the User Experience Questionnaire to evaluate participants' experiences with psilocybin and wearable device [23].

## 2.8. Fidelity

Quality assurance methods will be rigorously implemented throughout the study. A start-up meeting involving research coordinators, co-investigators, and partners will be held prior to the study launch. Comprehensive training will follow, emphasizing the study procedures and the completion of case report forms (CRFs). Those who are involved in administering the intervention in the present study will use checklists in order to ensure consistency in the delivery of the intervention materials. Additionally, 10% of clinical assessments recordings and CAPS-5 assessments will be reviewed by an independent rater to evaluate fidelity.

# 3. Outcomes

## 3.1. Primary outcome measure

Feasibility and tolerability will be determined using recruitment rates, withdrawal, data completion, adherence, and number and nature of AEs. We hypothesize that: 1) we will be able to recruit and assess 2–3 participants per month; 2) over 17 weeks of participation, the overall withdrawal rates will be no more than 20%; 3) the data completion rates will be more than 80%; 4) adherence rates, including treatment compliance and study completion, will be more than 80%; and 5) participants will report satisfaction with the content and delivery of the intervention, as measured by the exit survey and semi-structured qualitative interview. If any feasibility thresholds are not met, our team will analyze the underlying reasons and consider potential protocol modifications to address them.

## 3.2. Secondary outcome measure

The following domains will be assessed as secondary outcome measures:

- Clinician rated PTSD severity via CAPS-5 [24]

- Self-reported PTSD severity as measured with the Posttraumatic Stress Disorder Checklist for the DSM-5 (PCL-5) [25]

- Self-reported depression symptoms with the Patient Health Questionnaire-9 (PHQ-9) [26]

- Self-reported anxiety symptoms with the Generalized Anxiety Disorder Scale, 7-item (GAD-7) [27]

- Self-reported dissociation symptoms with the Dissociative Experiences Scale II (DES-II) [28]

- Self-reported sleep quality with the Pittsburgh Sleep Quality Index (PSQI) [29]

- Self-compassion, measured with the Self-Compassion Scale short form (SCS-SF) [30]

- Experiential avoidance, measured with the Brief Experiential Avoidance Questionnaire (BEAQ) [31]

- Relationship satisfaction, measured with the Quality of Relationships Inventory (QRI) [32]

- Functioning, psychological flexibility, emotional regulation, maladaptive beliefs, working alliance and well-being using: Inventory of Psychosocial Functioning (IPF), [33] 24 item Multidimensional Psychological Flexibility Inventory (MPFI-24), [34] Difficulties in Emotion Regulation Scale-Short Form (DERS-SF), [35] Posttraumatic Maladaptive Beliefs Scale (PMBS), [36] Working Alliance Inventory-Short Form (WAI-SF), [37] and 5-item World Health Organization Well-Being Index (WHO-5) [38]

### 3.3. Exploratory outcome

Digital physiological passive data collected through the use of a wearable device. Personal pDPP will be constructed based on the wearable and clinical assessment data.

   **3.3.1. Personal digital phenotype profile definition.**   We propose the definition of DPP, which calculates an individual's baseline and continuously monitors for changes in DPP characteristics derived from mathematical models. DPP is a representation of a user's physical and behavioral health using the baseline data collected at the beginning of the study as reference. The models are then fine-tuned with the continuous incoming data streams. However, variations of the model representations require further investigation to ensure its accuracy and robustness. This work investigates the statistical aspect of the analysis pipeline to offer a robust DPP representation. Specifically the use of robust principal component analysis (RPCA) to extract sparse representation. The DPP is developed through our Statistical, Information Theory, and Data-driven pipeline, then is represented through the sparse-rank matrix using RPCA. Once the DPP representation has been created, we additionally enhance this definition by proposing the term pDPP as a personalized version of the DPP, where we use collected longitudinal data to monitor the changes in physical and behavioral health of an individual participant.

### 3.4. Safety outcome measures

Safety measures will be applied to minimize risks associated with study participation. The safety of participants will be assured before, during and after the experimental session by assessing physiological effects, psychological distress, medical events, spontaneously reported reactions, and suicidality. Therapists and study physician will be available via mobile phone throughout the study to ensure participant safety. Vital Signs, Columbia Suicide Severity Rating Scale (C-SSRS), [39] AEs and spontaneously reported reactions will be systematically collected. AEs and SAEs will be documented from screening through study termination, and can be reported during in-person visits, over the phone, or via Zoom.

### 3.5. Data collection and clinical assessments

Assessments will occur 24 times throughout the study and will utilize various clinical scales (CAPS-5, PCL-5, PHQ-9, GAD-7, DES-II, PSQI, QRI, WHO-5, IPF, PMBS, BEAQ, MPFI-24,

WAI-SF, DERS-SF, SCF-SF, PIQ, CTQ-SF, EBI, ASC, APEQ) to measure symptoms severity and treatment efficacy. The CAPS-5 will be assessed at baseline, day 7, week 2, week 5 and week 13 follow-ups. PCL-5, PHQ-9, GAD-7 and WHO-5 will be measured at baseline, pre-baseline wearable monitoring, daily during the intervention week and follow-ups visits. The WAI-SF will be administered daily throughout the intervention week. Furthermore, we will administer DES-II, PSQI, QRI, WHO-5, IPF, PMBS, BEAQ, MPFI-24, DERS-SF and SCF-SF at baseline, day 7 and week 5 and week 13 follow-up timepoints. The PIQ, EBI, ASC and APEQ will be administered post-psilocybin dosing to evaluate psychedelic experience. All participants will be screened at least four weeks prior to the first therapy session. Study personnel will input data into REDCap, a secure data collecting tool. Only authorized research workers will be able to access the data, which will be kept on secure cloud servers.

## 3.6. Confidentiality

The confidentiality of the data collected and the identity of the individuals participating in this study will be strictly maintained. All files pertaining to subjects in the study will be assigned a unique ID. Personal identifiable information (PII), such as name, address, telephone number, email address, and date-of-birth, will be restricted from access by anyone not authorized to conduct the study. Only the research coordinator responsible for direct communications with participants will have access to PII. Source documents will be securely stored in a locked filing cabinet to limit access, while electronic source documents will be password-protected and saved on a secure server. Importantly, our CRFs will not contain any personal health information; only a unique ID will be recorded. If a participant's name appears on any other document (e.g., laboratory report), it will be de-identified and replaced with the unique ID on copies retained in the Trial Master File or available for audit.

Study findings stored on a computer will comply with Unity Health Toronto's data and privacy regulations. Access to all source documents collected during the study will be granted to the research coordinator, research assistants, Principal Investigator and co-investigators. Participants will be informed that representatives from other parties including pharmaceutical companies, the Research Ethics Board (REB), and regulatory authorities may inspect their records. All information will be handled with the utmost confidentiality and in accordance with Unity Health Toronto's data and privacy policies. The investigator will maintain a personal subject identification list linking unique IDs to participant names, facilitating.

Data collected from the wearable device will be de-identified on the device's platform. Each participant's account will be created using a dummy email address and password linked to a unique ID, thereby safeguarding participant privacy After de-identification, the data will be transferred to an external encrypted and password-protected hard drive. During data analysis, only de-identified data will be provided to the analysts. This will be achieved using a two-zone approach. This process will utilize a two-zone approach, which divides study team members into two groups; identified and de-identified. The identified group will have access to both identified (i.e., demographic) and de-identified data, enabling them to perform data management, quality control, and data linkage functions for both active and passive data. In contrast, the de-identified group will have access solely to the de-identified information. This structure ensures that data analysts only access de-identified data, protecting participant privacy and preventing access to any identifying information.

## 3.7. Informed consent

Prior to participation, the study coordinator will engage with potential participants to provide detailed information about the study's objectives and the consent document. Participants will

be given adequate time to review the informed consent form and ask any questions for clarification before making their decision. Those who choose to participate will be asked to sign the consent form.

### 3.8. Sample size

Due to the unknown anticipated effect size, a formal sample size calculation cannot be conducted. However, we expect to recruit a total of 15 participants for this trial.

### 3.9. Monitoring

A data monitoring committee will not be established due to the small sample size and the specific objectives of the trial. Additionally, there will be no pre-specified independent audit.

## 4. Statistical analysis

Feasibility outcomes will be reported with descriptive statistics (i.e., counts and proportions). A minimum of 2–3 participants per month will be considered a sufficient recruitment rate to indicate the feasibility of a future larger multi-site trial. We will estimate the proportions of participants adherent with the protocol, that withdraw, and data completion with a 95% CI. A full trial with this design will be deemed feasible if the lower 95% confidence limit for the rates of adherence and data completion are $\geq 80\%$, and if the upper limit of 95% confidence interval for the withdrawal rate is $\leq 20\%$. Frequencies and proportions of AEs and serious AEs will be tabulated for the total sample for each AE type. For continuous outcomes, these measures are ordinal in nature. Therefore, non-parametric methods will be used to analyze changes in scores. Means, standard deviations, and 95% CIs will be reported for descriptive purposes, but the primary analysis will involve non-parametric tests to compare changes in scores. To assess changes in PTSD symptoms, a Wilcoxon signed-rank test will be used to compare CAPS-5 scores from baseline until the end of treatment and over the follow-up visits. We will estimate the standard deviation of PTSD severity (CAPS-5) and its within person correlation between the baseline and the end of treatment scores. The study will examine the effect size associated with psilocybin-assisted CPT in participants with PTSD, including the durability of effects on PTSD symptoms. The main analysis will be a comparison of PTSD symptoms before and after treatment. There will be analyses of changes in symptoms of depression, psychological functioning, sleep quality and PTSD severity. One week following the end of treatment, the primary endpoint assessment will take place for all subjects. An independent rater not involved in therapy will administer the CAPS-5. C-SSRS ratings will be completed by study coordinator, and other measures will be based on subjects' self-report. Baseline will be compared to the primary end point and follow up time points. This study will provide data on the impact of psilocybin in conjunction with massed CPT. Based on the outcomes, this study will primarily detect effects due to psilocybin-assisted CPT on PTSD symptoms through calculating effect sizes and significance testing. A Wilcoxon signed rank test comparing CAPS-5 scores from the baseline to the primary endpoint will be the main analysis. Durability of effects will be examined by comparing changes from baseline to the assessments at 1 until 12 weeks after the final CPT session (day 7) for each PTSD subject using Friedman test, which is a non-parametric test method for repeated measures. Effects on depression symptoms, psychological functioning, trauma-related beliefs and sleep quality will also be assessed for exploratory purposes. Changes in these measures will be compared from baseline to primary endpoint for participants using Wilcoxon signed rank tests. Analyses will examine changes from baseline up to week 13 using Friedman tests. Effect sizes for the psilocybin-assisted CPT will be calculated at the primary endpoint and follow-up time points.

Passive data is effective at identifying behaviors and trends in activity but is poor in measuring people's internal states, motivation, and attitude, whereas active data is the opposite. Developing a methodology for integrating the two data sources can mitigate their respective weaknesses. The passive data capturing activity, sleep and physiological data collected by the wearable device will be correlated and fused with the active data collected by the clinical data. The motivation to fuse active and passive data includes the cross-validation and improvement of measurements, the explanation of human behavior, and novel opportunities to improve representative models in experimental settings. Techniques such as feature extraction, dimension reduction, feature relevance estimation for ranking, statistical analysis, and data fusion will be used on the passive and active.

The active data will serve as a ground truth to validate the passive data from the wearable device for monitoring anxiety and depression. We will use generalized machine learning models to detect and predict distress among participants and develop personalized time series machine learning models for personalized application. This will involve creating predictive models based on a participant's data and implementing the model suited for the participant. This proposal will be an iterative process. To mitigate missing data, suitable filtering and encoders may be used to interpolate and predict the expected data accordingly. During the analysis, we will extract features from the passive physiological data for better representation. Trends, classifications, and relationships will be determined using different machine learning techniques. Implementations of pre-processing, feature extraction and machine learning will be conducted in a numerical program for efficient and powerful processing that will achieve impactful results.

## 5. Reporting

Outcomes will be reported in accordance with CONSORT guidelines. [40] Following the completion of the study, the results will undergo peer review for publication in a respected academic journal and will also be presented at esteemed scientific conferences. The authors are dedicated to upholding the integrity of the publication process and will not employ professional writers.

## 6. Discussion

This open-label trial serves as an essential first step toward gathering precise estimates of feasibility, tolerability, efficacy, and safety outcome variability in psilocybin-assisted massed CPT for chronic PTSD. By utilizing multiple clinical scales—ranging from symptom severity to functional outcomes—this study offers a detailed view of patient responses, which will be instrumental in the design and planning of a larger, sufficiently powered RCT. Given the disabling nature of PTSD and the need for rapid-acting, more effective treatments, these findings could help refine future interventions and improve front-line therapeutic options. The trial is aligned with standard medical research methodologies, ensuring that the subsequent RCT is rigorously designed and adequately powered to provide reliable and clinically meaningful outcomes for individuals suffering from chronic PTSD.

## Supporting information

**S1 File.**
(DOCX)

**S2 File.**
(DOCX)

**S3 File. SPIRIT 2013 checklist: Recommended items to address in a clinical trial protocol and related documents\*.**
(PDF)

## Author Contributions

**Conceptualization:** Shakila Meshkat, Kathleen Stewart, Venkat Bhat.

**Investigation:** Shakila Meshkat, Richard J. Zeifman, Kathleen Stewart, Reinhard Janssen-Aguilar, Wendy Lou, Rakesh Jetly, Venkat Bhat.

**Methodology:** Shakila Meshkat, Richard J. Zeifman, Kathleen Stewart, Wendy Lou, Rakesh Jetly, Venkat Bhat.

**Project administration:** Shakila Meshkat, Richard J. Zeifman, Kathleen Stewart, Reinhard Janssen-Aguilar, Wendy Lou, Rakesh Jetly, Candice M. Monson, Venkat Bhat.

**Resources:** Shakila Meshkat, Venkat Bhat.

**Supervision:** Venkat Bhat.

**Validation:** Shakila Meshkat, Richard J. Zeifman, Kathleen Stewart, Venkat Bhat.

**Visualization:** Shakila Meshkat, Richard J. Zeifman, Kathleen Stewart, Venkat Bhat.

**Writing – original draft:** Shakila Meshkat, Richard J. Zeifman, Kathleen Stewart, Reinhard Janssen-Aguilar, Venkat Bhat.

**Writing – review & editing:** Shakila Meshkat, Richard J. Zeifman, Kathleen Stewart, Reinhard Janssen-Aguilar, Wendy Lou, Rakesh Jetly, Candice M. Monson, Venkat Bhat.

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
