## [Decision Letter · Decision Letter 0]

9 Sep 2024

PONE-D-24-19436Psilocybin-Assisted Massed Cognitive Processing Therapy for Chronic Posttraumatic Stress Disorder: Protocol for an Open-Label TrialPLOS ONE

Dear Dr. Bhat,

Thank you for submitting your manuscript to PLOS ONE. After careful consideration, we feel that it has merit but does not fully meet PLOS ONE’s publication criteria as it currently stands. Therefore, we invite you to submit a revised version of the manuscript that addresses the points raised during the review process.

We look forward to receiving your revised manuscript.

Kind regards,

Domenico Fuoco

Academic Editor

PLOS ONE

Journal Requirements:

"I have read the journal's policy and the authors of this manuscript have the following competing interests: VB is supported by an Academic Scholar Award from the University of Toronto Department of Psychiatry and has received research support from the Canadian Institutes of Health Research, Brain & Behavior Foundation, Ontario Ministry of Health Innovation Funds, Royal College of Physicians and Surgeons of Canada, Department of National Defense (Government of Canada), New Frontiers in Research Fund, Associated Medical Services Inc. Healthcare, American Foundation for Suicide Prevention, Roche Canada, Novartis, and Eisai. RJ is the CMO of Mydecine Innovation Group."

7. We note that the original protocol file you uploaded contains a confidentiality notice indicating that the protocol may not be shared publicly or be published. Please note, however, that the PLOS Editorial Policy requires that the original protocol be published alongside your manuscript in the event of acceptance. Please note that should your paper be accepted, all content including the protocol will be published under the Creative Commons Attribution (CC BY) 4.0 license, which means that it will be freely available online, and any third party is permitted to access, download, copy, distribute, and use these materials in any way, even commercially, with proper attribution.

Therefore, we ask that you please seek permission from the study sponsor or body imposing the restriction on sharing this document to publish this protocol under CC BY 4.0 if your work is accepted. We kindly ask that you upload a formal statement signed by an institutional representative clarifying whether you will be able to comply with this policy. Additionally, please upload a clean copy of the protocol with the confidentiality notice (and any copyrighted institutional logos or signatures) removed.

Additional Editor Comments:

I agree on the pivotal importance of your work and I fully endorse your manuscript. Please, as per the reviewer #1 request, provide in your test a more specific details on the nature of your analysis vs review. This very simple comment that I am asking you to add, it will make your manuscript perfect.

I am also very sorry for the long time taken in this peer-review process.

Feel free to contact me anytime.

A+

Reviewers' comments:

Reviewer's Responses to Questions

**Comments to the Author**

1. Does the manuscript provide a valid rationale for the proposed study, with clearly identified and justified research questions?

Reviewer #1: Yes

Reviewer #2: Yes

2. Is the protocol technically sound and planned in a manner that will lead to a meaningful outcome and allow testing the stated hypotheses?

Reviewer #1: Yes

Reviewer #2: Yes

3. Is the methodology feasible and described in sufficient detail to allow the work to be replicable?

Reviewer #1: Yes

Reviewer #2: Yes

4. Have the authors described where all data underlying the findings will be made available when the study is complete?

Reviewer #1: Yes

Reviewer #2: Yes

5. Is the manuscript presented in an intelligible fashion and written in standard English?

Reviewer #1: Yes

Reviewer #2: Yes

6. Review Comments to the Author

You may also provide optional suggestions and comments to authors that they might find helpful in planning their study.

Reviewer #1: Important note: This review pertains only to ‘statistical aspects’ of the study and so ‘clinical aspects’ [like medical importance, relevance of the study, ‘clinical significance and implication(s)’ of the whole study, etc.] are to be evaluated [should be assessed] separately/independently. Further please note that any ‘statistical review’ is generally done under the assumption that study specific methodological [as well as execution] issues are perfectly taken care of by the investigator(s). This review is not an exception to that and so does not cover clinical aspects {however, seldom comments are made only if those issues are intimately / scientifically related & intermingle with ‘statistical aspects’ of the study}. Agreed that ‘statistical methods’ are used as just tools here, however, they are vital part of methodology [and so should be given due importance]. I look at the manuscript in/with statistical view point, other reviewer(s) look(s) at it with different angle so that in totality the review is very comprehensive. However, there should be efforts from authors side to improve (may be by taking clues from reviewer’s comments). Therefore, please do not limit the revision only (with respect) to comments made here.

COMMENTS: Although there is no major flaw [study/manuscript is already of acceptable level], I have different opinion / observations/concerns regarding (very) few issues which are given below, if agreed, try to incorporate them (as this study is at planning/protocol stage only)]:

In ‘Abstract-Background’ section it is stated that “This open-label pilot study will examine the feasibility, tolerability, and preliminary efficacy of a single dose 25 mg psilocybin, combined with 1 week of massed CPT in chronic PTSD”. However, in the title “pilot / feasibility study” is not mentioned. Why? You may know that many things are often ignored (loosely looked at / evaluated) in case of ‘pilot studies’ [typically involving a small number of subjects], and that more latitude [i.e., leeway, freedom, liberty] in statistical requirements is observed/given. Nevertheless, pilot nature of the study is expected to be mentioned in the title itself.

For a pilot study it is alright to have ‘single-arm design’, [or it is alright when that is the only possibility’], however, it is very essential to keep the limitations in mind while interpreting results. This study being ‘pilot’ (feasibility study) in nature, sample size is not a big issue. However, few important methodological issues need to be very rigorous (to the maximum possible extent) followed. Very good that in section ‘5. Reporting’ you stated “Outcomes will be reported according to CONSORT reporting guidelines” and quoted reference 39 [Schulz KF, Altman DG, Moher D, CONSORT Group. CONSORT 2010 Statement: updated guidelines for reporting parallel group randomised trials. BMC Med 2010; 8: 18], however, please note that this being a ‘single-arm design’ pilot study many items are not applicable {and so keep the limitations in mind}. You may be aware of one important reference, namely: “Introducing the CONSORT extension to pilot trials: enhancing the design, conduct and reporting of pilot or feasibility trials” by Luciana P. F. Abbade, Joelcio F. Abbade and Lehana Thabane.

As said in section ‘7. Discussion’ [This open label trial is primarily interested in precise estimates of feasibility, tolerability, efficacy and safety outcome variability, in patients with chronic PTSD that will aid in the planning of a larger, sufficiently powered RCT. PTSD is a disabling and severe psychiatric disorder, and it is imperative to develop rapid-acting interventions and increase the efficacy of current front-line treatments. This open-label trial is a first step towards what may be a promising intervention for chronic PTSD. Findings from this study may help to develop psilocybin-assisted iCPT as a novel and rapid acting intervention for individuals suffering with chronic PTSD] and because you intend to use many scales [Assessments will consist of several clinical scales (CAPS-5, PCL-5, PHQ-9, GAD-7, DES-II, PSQI, QRI, WHO-5, IPF, PMBS, BEAQ, MPFI-24, WAI-SF, DERS-SF, SCF-SF, PIQ, CTQ-SF, EBI, ASC, APEQ) to measure symptoms severity and efficacy], by rightly guessing that you all (team) are likely be involved in the planning of a larger, sufficiently powered RCT subsequently and in this context, I request all authors to read the following note pasted from one standard textbook on ‘Medical Research Methodology’ [though I am sure that the authors already know these things]:

Though the measures/tools used are appropriate often times, most of them are likely to yield data that are in [at the most] ‘ordinal’ level of measurement [and not in ratio level of measurement for sure {as the score two times higher does not indicate presence of that parameter/phenomenon as double (for example, a Visual Analogue Scales VAS score or say ‘depression’ score)}]. Then application of suitable non-parametric (or distribution free) test(s) is/are indicated/advisable [even if distribution may be ‘Gaussian’ (also called ‘normal’)]. Agreed that there is/are no non-parametric test(s)/technique(s) available to be used as alternative in all situation(s), but should be used whenever/wherever they are available. Therefore, in short use suitable non-parametric test(s)/technique(s) while dealing with data that are in ‘ordinal’ level of measurement even if [despite that] the distribution may be ‘Gaussian’. Testing ‘normality’ in sample [by using any normality test(s)} is not required/desired while dealing with data that are in ‘ordinal’ level of measurement [as most of the normality tests are not valid for ‘ordinal’ data].

In section ‘4. Statistical analysis’ you have mentioned ‘non-parametric’ test [Wilcoxon signed rank test comparing CAPS-5 scores from the baseline to the primary endpoint will be the main analysis] which shows that right things are known to the team and so this a sort of reminder to keep mind.

As pointed out in ‘important note’ above “This review pertains only to ‘statistical aspects’ of the study and so ‘clinical aspects’ should be assessed separately/independently. In my opinion, to make this article finally acceptable (which is quite possible and easy), a small amount of re-vision (re-drafting) may be needed. ‘Minor revision’ is recommended.

Reviewer #2: The abstract and introduction both outlines clearly the question of using Cognitive Processing Therapy (CPT), with psychedelic substances as an adjunct, in the treatment of chronic PTSD. The point is emphasized upon with a solid foundational background, offering context and supporting the question in focus. The points of analysis—feasibility, tolerability, and preliminary efficacy of the treatment—are clearly stated. As PTSD and psychedelics is an ongoing area of research, this specific question has the potential for valuable insights. Using specific modes of therapy may offer new avenues for PTSD/psychedelics research, as well.

The methodology and design of the trial is well-planned, with its primary, secondary, and tertiary objectives clearly stated. Safety objectives are also touched upon. The use of strict inclusion/exclusion criteria, as well as a pre-screening session for potential subjects ensures that eligibility criteria are met and that subjects are truly suitable. This gives the results true relevance and statistical power, with the outcomes measures used. Those using psychotropic medications will be required to taper, removing confounding factors from the analysis. I suggest the authors use a greater sample size than n=15, to give greater statistical power for gathered data.

The treatment section describes in sufficient detail the use of psilocybin in combination with psychotherapy sessions given at regular intervals, plus the use of a wearable device to passively gather physiological data through the duration. Mentioning specifically the treatment regimen for psilocybin administration, the schedule for psychotherapy sessions, and the timeframe for use of the wearable device allows this experimental procedure to be reliably replicated. The use of their Digital Phenotype Profile Definition (DPP) also serves as another means of replicability, relying on mathematical models based on continuously incoming subject data. The schema and calendars offered also adequately illustrates the study design and schedule.

In the abstract, the authors clearly state that the data analyzed at the conclusion of this trial will be used in the design and execution of a “multi-site, large-scale randomized control trial to assess the efficacy of psilocybin-assisted CPT for PTSD”; I believe this implies the data gathered will be preliminary and used to guide subsequent trials exploring this question. Beyond this, exactly where/how data will be made publicly available is not mentioned. The authors may consider referencing this preliminary research in the prospective manuscript detailing the larger-scale trial, come that time.

This manuscript is well-written, with grammatically correct English and proper diction and syntax. Their ideas, methods, and rationale are elaborated upon with adequate depth, and each aspect of the trial is clearly expressed. I believe the extensive background the authors give on both psychological and physiological aspects, and how these will be accounted for throughout the trial will help readers fully grasp the questions this research is trying to answer, and how the researchers will go about answering them. I believe this trial has the potential to offer insight into important questions, offering a more nuanced analysis of psychedelics as an adjunct in the treatment of PTSD. I would suggest this trial be executed and the data analyzed, published, and used as a framework for the larger scale work the authors are planning.

7. PLOS authors have the option to publish the peer review history of their article (what does this mean?). If published, this will include your full peer review and any attached files.

Reviewer #1: No

Reviewer #2: **Yes: **Christopher Matthew Teske

---

## [Author Response · Author response to Decision Letter 0]

4 Oct 2024

Reviewer #1: Important note: This review pertains only to ‘statistical aspects’ of the study and so ‘clinical aspects’ [like medical importance, relevance of the study, ‘clinical significance and implication(s)’ of the whole study, etc.] are to be evaluated [should be assessed] separately/independently. Further please note that any ‘statistical review’ is generally done under the assumption that study specific methodological [as well as execution] issues are perfectly taken care of by the investigator(s). This review is not an exception to that and so does not cover clinical aspects {however, seldom comments are made only if those issues are intimately / scientifically related & intermingle with ‘statistical aspects’ of the study}. Agreed that ‘statistical methods’ are used as just tools here, however, they are vital part of methodology [and so should be given due importance]. I look at the manuscript in/with statistical view point, other reviewer(s) look(s) at it with different angle so that in totality the review is very comprehensive. However, there should be efforts from authors side to improve (may be by taking clues from reviewer’s comments). Therefore, please do not limit the revision only (with respect) to comments made here.

Response: We would like to thank you for your thorough review and valuable feedback regarding the statistical aspects of our manuscript. Based on your comments, we have revised the statistical section of the manuscript for clarity.

COMMENTS: Although there is no major flaw [study/manuscript is already of acceptable level], I have different opinion / observations/concerns regarding (very) few issues which are given below, if agreed, try to incorporate them (as this study is at planning/protocol stage only)]:

In ‘Abstract-Background’ section it is stated that “This open-label pilot study will examine the feasibility, tolerability, and preliminary efficacy of a single dose 25 mg psilocybin, combined with 1 week of massed CPT in chronic PTSD”. However, in the title “pilot / feasibility study” is not mentioned. Why? You may know that many things are often ignored (loosely looked at / evaluated) in case of ‘pilot studies’ [typically involving a small number of subjects], and that more latitude [i.e., leeway, freedom, liberty] in statistical requirements is observed/given. Nevertheless, pilot nature of the study is expected to be mentioned in the title itself.

Response: We appreciate your feedback. Based on your comment, we received the manuscript title: ‘’Psilocybin-Assisted Massed Cognitive Processing Therapy for Chronic Posttraumatic Stress Disorder: Protocol for an Open-Label Pilot Feasibility Trial.’’

For a pilot study it is alright to have ‘single-arm design’, [or it is alright when that is the only possibility’], however, it is very essential to keep the limitations in mind while interpreting results. This study being ‘pilot’ (feasibility study) in nature, sample size is not a big issue. However, few important methodological issues need to be very rigorous (to the maximum possible extent) followed. Very good that in section ‘5. Reporting’ you stated “Outcomes will be reported according to CONSORT reporting guidelines” and quoted reference 39 [Schulz KF, Altman DG, Moher D, CONSORT Group. CONSORT 2010 Statement: updated guidelines for reporting parallel group randomised trials. BMC Med 2010; 8: 18], however, please note that this being a ‘single-arm design’ pilot study many items are not applicable {and so keep the limitations in mind}. You may be aware of one important reference, namely: “Introducing the CONSORT extension to pilot trials: enhancing the design, conduct and reporting of pilot or feasibility trials” by Luciana P. F. Abbade, Joelcio F. Abbade and Lehana Thabane.

Response: We thank the reviewer for their comment. We changed the reference and references Luciana et al. article in the revised manuscript.

As said in section ‘7. Discussion’ [This open label trial is primarily interested in precise estimates of feasibility, tolerability, efficacy and safety outcome variability, in patients with chronic PTSD that will aid in the planning of a larger, sufficiently powered RCT. PTSD is a disabling and severe psychiatric disorder, and it is imperative to develop rapid-acting interventions and increase the efficacy of current front-line treatments. This open-label trial is a first step towards what may be a promising intervention for chronic PTSD. Findings from this study may help to develop psilocybin-assisted iCPT as a novel and rapid acting intervention for individuals suffering with chronic PTSD] and because you intend to use many scales [Assessments will consist of several clinical scales (CAPS-5, PCL-5, PHQ-9, GAD-7, DES-II, PSQI, QRI, WHO-5, IPF, PMBS, BEAQ, MPFI-24, WAI-SF, DERS-SF, SCF-SF, PIQ, CTQ-SF, EBI, ASC, APEQ) to measure symptoms severity and efficacy], by rightly guessing that you all (team) are likely be involved in the planning of a larger, sufficiently powered RCT subsequently and in this context, I request all authors to read the following note pasted from one standard textbook on ‘Medical Research Methodology’ [though I am sure that the authors already know these things]:

Response: We thank the reviewer for their comment. Based on your comment, we revised the discussion section: ‘’This open-label trial serves as an essential first step toward gathering precise estimates of feasibility, tolerability, efficacy, and safety outcome variability in psilocybin-assisted massed CPT for chronic PTSD. By utilizing multiple clinical scales—ranging from symptom severity to functional outcomes—this study offers a detailed view of patient responses, which will be instrumental in the design and planning of a larger, sufficiently powered RCT. Given the disabling nature of PTSD and the need for rapid-acting, more effective treatments, these findings could help refine future interventions and improve front-line therapeutic options. The trial is aligned with standard medical research methodologies, ensuring that the subsequent RCT is rigorously designed and adequately powered to provide reliable and clinically meaningful outcomes for individuals suffering from chronic PTSD.’’

Though the measures/tools used are appropriate often times, most of them are likely to yield data that are in [at the most] ‘ordinal’ level of measurement [and not in ratio level of measurement for sure {as the score two times higher does not indicate presence of that parameter/phenomenon as double (for example, a Visual Analogue Scales VAS score or say ‘depression’ score)}]. Then application of suitable non-parametric (or distribution free) test(s) is/are indicated/advisable [even if distribution may be ‘Gaussian’ (also called ‘normal’)]. Agreed that there is/are no non-parametric test(s)/technique(s) available to be used as alternative in all situation(s), but should be used whenever/wherever they are available. Therefore, in short use suitable non-parametric test(s)/technique(s) while dealing with data that are in ‘ordinal’ level of measurement even if [despite that] the distribution may be ‘Gaussian’. Testing ‘normality’ in sample [by using any normality test(s)} is not required/desired while dealing with data that are in ‘ordinal’ level of measurement [as most of the normality tests are not valid for ‘ordinal’ data].

In section ‘4. Statistical analysis’ you have mentioned ‘non-parametric’ test [Wilcoxon signed rank test comparing CAPS-5 scores from the baseline to the primary endpoint will be the main analysis] which shows that right things are known to the team and so this a sort of reminder to keep mind.

Response: We thank the reviewer for their comment. We’ve revised the Statistical analysis section: “For continuous outcomes, these measures are ordinal in nature. Therefore, non-parametric methods will be used to analyze changes in scores. Means, standard deviations, and 95% CIs will be reported for descriptive purposes, but the primary analysis will involve non-parametric tests to compare changes in scores. To assess changes in PTSD symptoms, a Wilcoxon signed-rank test will be used to compare CAPS-5 scores from baseline until the end of treatment and over the follow-up visits.”

Reviewer #2: The abstract and introduction both outlines clearly the question of using Cognitive Processing Therapy (CPT), with psychedelic substances as an adjunct, in the treatment of chronic PTSD. The point is emphasized upon with a solid foundational background, offering context and supporting the question in focus. The points of analysis—feasibility, tolerability, and preliminary efficacy of the treatment—are clearly stated. As PTSD and psychedelics is an ongoing area of research, this specific question has the potential for valuable insights. Using specific modes of therapy may offer new avenues for PTSD/psychedelics research, as well.

The methodology and design of the trial is well-planned, with its primary, secondary, and tertiary objectives clearly stated. Safety objectives are also touched upon. The use of strict inclusion/exclusion criteria, as well as a pre-screening session for potential subjects ensures that eligibility criteria are met and that subjects are truly suitable. This gives the results true relevance and statistical power, with the outcomes measures used. Those using psychotropic medications will be required to taper, removing confounding factors from the analysis. I suggest the authors use a greater sample size than n=15, to give greater statistical power for gathered data.

Response: We thank the reviewer for their comment. We acknowledge the suggestion to increase the number of participants to improve statistical power. However, we would like to note that the current sample size is based on careful consideration of the early-phase nature of this trial. Given that this is a pilot study, our primary goal is to assess feasibility, safety, and preliminary efficacy, which will inform the design of a larger-scale randomized controlled trial in the future. The selected sample size is sufficient for these early objectives and is also aligned with logistical and resource constraints. We appreciate the feedback and will consider increasing the sample size in subsequent studies.

The treatment section describes in sufficient detail the use of psilocybin in combination with psychotherapy sessions given at regular intervals, plus the use of a wearable device to passively gather physiological data through the duration. Mentioning specifically the treatment regimen for psilocybin administration, the schedule for psychotherapy sessions, and the timeframe for use of the wearable device allows this experimental procedure to be reliably replicated. The use of their Digital Phenotype Profile Definition (DPP) also serves as another means of replicability, relying on mathematical models based on continuously incoming subject data. The schema and calendars offered also adequately illustrates the study design and schedule.

In the abstract, the authors clearly state that the data analyzed at the conclusion of this trial will be used in the design and execution of a “multi-site, large-scale randomized control trial to assess the efficacy of psilocybin-assisted CPT for PTSD”; I believe this implies the data gathered will be preliminary and used to guide subsequent trials exploring this question. Beyond this, exactly where/how data will be made publicly available is not mentioned. The authors may consider referencing this preliminary research in the prospective manuscript detailing the larger-scale trial, come that time.

Response: Thank you for your comment. Based on your comment, we revised the abstract section: ‘’Background: Posttraumatic stress disorder (PTSD) affects 3.9% of the general population. While massed cognitive processing therapy (CPT) has demonstrated efficacy in treating chronic PTSD, a substantial proportion of patients still continue to meet PTSD criteria after treatment, highlighting the need for novel therapeutic approaches. Preliminary evidence supports the potential therapeutic action of psilocybin to alleviate PTSD symptoms. This open-label pilot study aims to evaluate the feasibility, tolerability, and preliminary efficacy of a single dose 25 mg psilocybin in combination with one week of massed CPT in patients with chronic PTSD. Method: Fifteen participants with chronic PTSD will undergo 12 CPT sessions, two psilocybin-related psychotherapy sessions, and one psilocybin dosing session over a 7-days period. The primary outcomes are feasibility and tolerability, which will be measured by recruitment rates, withdrawal, data completion, adherence, number and nature of adverse events. Secondary objectives include assessing the preliminary efficacy of psilocybin-assisted CPT in reducing PTSD severity, self-reported treatment outcomes and exploring putative mechanisms of change. Participants will be monitored weekly for 12 weeks post-treatment and passive data relevant to mental health and well-being will be collected using a wearable device. Discussion: This trial will generate important preliminary data on the use of psilocybin-assisted CPT for treating PTSD. The findings will guide the design of a multi-site, large-scale randomized control trial to more rigorously assess the efficacy of this intervention. De-identified data from this study will be available upon request after publication of the results. This study represents a promising and innovative approach to PTSD treatment, potentially offering an alternative therapeutic option for individuals unresponsive to conventional therapies.’’

This manuscript is well-written, with grammatically correct English and proper diction and syntax. Their ideas, methods, and rationale are elaborated upon with adequate depth, and each aspect of the trial is clearly expressed. I believe the extensive background the authors give on both psychological and physiological aspects, and how these will be accounted for throughout the trial will help readers fully grasp the questions this research is trying to answer, and how the researchers will go about answering them. I believe this trial has the potential to offer insight into important questions, offering a more nuanced analysis of psychedelics as an adjunct in the treatment of PTSD. I would suggest this trial be executed and the data analyzed, published, and used as a framework for the larger scale work the authors are planning.

Response: Thank you for your feedback on our manuscript. We are pleased that you found the writing clear and the trial design well-explained, particularly in addressing both the psychological and physiological aspects of the study. Your suggestion that this trial could serve as a framework for larger-scale research aligns with our long-term goals, and we look forward to building on this work in the future.

---

## [Decision Letter · Decision Letter 1]

31 Oct 2024

Psilocybin-Assisted Massed Cognitive Processing Therapy for Chronic Posttraumatic Stress Disorder: Protocol for an Open-Label Pilot Feasibility Trial

PONE-D-24-19436R1

Dear Dr. Bhat,

We’re pleased to inform you that your manuscript has been judged scientifically suitable for publication and will be formally accepted for publication once it meets all outstanding technical requirements.

Kind regards,

Jan Christopher Cwik, Ph.D.

Academic Editor

PLOS ONE

Additional Editor Comments (optional):

Reviewers' comments:

Reviewer's Responses to Questions

**Comments to the Author**

1. Does the manuscript provide a valid rationale for the proposed study, with clearly identified and justified research questions?

Reviewer #1: Yes

Reviewer #2: Yes

2. Is the protocol technically sound and planned in a manner that will lead to a meaningful outcome and allow testing the stated hypotheses?

Reviewer #1: Yes

Reviewer #2: Yes

3. Is the methodology feasible and described in sufficient detail to allow the work to be replicable?

Reviewer #1: Yes

Reviewer #2: Yes

4. Have the authors described where all data underlying the findings will be made available when the study is complete?

Reviewer #1: Yes

Reviewer #2: Yes

5. Is the manuscript presented in an intelligible fashion and written in standard English?

Reviewer #1: Yes

Reviewer #2: Yes

6. Review Comments to the Author

You may also provide optional suggestions and comments to authors that they might find helpful in planning their study.

Reviewer #1: COMMENTS: I am very happy to note that all of the comments made on earlier draft are/were considered positively & are attended satisfactorily, I recommend the acceptance without any hesitation [rather happily now].

Reviewer #2: The authors have adequately addressed the initial concerns of the reviewers. After careful review of the revised version, I beleive this manuscript is now suitable for publication.

7. PLOS authors have the option to publish the peer review history of their article (what does this mean?). If published, this will include your full peer review and any attached files.

Reviewer #1: No

Reviewer #2: **Yes: **Christopher Matthew Teske

---

## [Editor Report · Acceptance letter]

4 Nov 2024

PONE-D-24-19436R1 

PLOS ONE

Dear Dr. Bhat, 

I'm pleased to inform you that your manuscript has been deemed suitable for publication in PLOS ONE. Congratulations! Your manuscript is now being handed over to our production team.

Kind regards, 

on behalf of

Dr. Jan Christopher Cwik 

Academic Editor

PLOS ONE